# Ammonia Production by *Streptomyces* Symbionts of *Acromyrmex* Leaf-Cutting Ants Strongly Inhibits the Fungal Pathogen *Escovopsis*

**DOI:** 10.3390/microorganisms9081622

**Published:** 2021-07-29

**Authors:** Basanta Dhodary, Dieter Spiteller

**Affiliations:** Chemical Ecology/Biological Chemistry, University of Konstanz, Universitätsstrasse 10, 78457 Konstanz, Germany; basanta02@hotmail.com

**Keywords:** antifungal, bioassay, chemical ecology, closed loop stripping, volatiles

## Abstract

Leaf-cutting ants live in mutualistic symbiosis with their garden fungus *Leucoagaricus gongylophorus* that can be attacked by the specialized pathogenic fungus *Escovopsis*. *Actinomyces* symbionts from *Acromyrmex* leaf-cutting ants contribute to protect *L. gongylophorus* against pathogens. The symbiont *Streptomyces* sp. Av25_4 exhibited strong activity against *Escovopsis weberi* in co-cultivation assays. Experiments physically separating *E. weberi* and *Streptomyces* sp. Av25_4 allowing only exchange of volatiles revealed that *Streptomyces* sp. Av25_4 produces a volatile antifungal. Volatile compounds from *Streptomyces* sp. Av25_4 were collected by closed loop stripping. Analysis by NMR revealed that *Streptomyces* sp. Av25_4 overproduces ammonia (up to 8 mM) which completely inhibited the growth of *E. weberi* due to its strong basic pH. Additionally, other symbionts from different *Acromyrmex* ants inhibited *E. weberi* by production of ammonia. The waste of ca. one third of *Acomyrmex* and *Atta* leaf-cutting ant colonies was strongly basic due to ammonia (up to ca. 8 mM) suggesting its role in nest hygiene. Not only complex and metabolically costly secondary metabolites, such as polyketides, but simple ammonia released by symbionts of leaf-cutting ants can contribute to control the growth of *Escovopsis* that is sensitive to ammonia in contrast to the garden fungus *L. gongylophorus*.

## 1. Introduction

Leaf-cutting ants belonging to the genus *Acromyrmex* or *Atta* (Hymenoptera: Formicidae: Myrmicinae), are particular among ants because they cultivate the fungus *Leucoagaricus gongylophorus* (Möller, 1893, Agaricales: Agaricaceae) in designated chambers of their nests. Leaf-cutting ants live in mutualistic symbiosis with their garden fungus. The ants cut plant material and bring it to their nests where ant workers process the plant material into small pieces and inoculate their garden fungus with them. The ants perform this fungiculture because they depend on *L. gongylophorus* as food source [1,2]. *L. gongylophorus* can efficiently degrade the plant material and thus makes the nutrients from the plant tissue available for the leaf-cutting ants [3,4]. In addition to the supply with plant material, leaf-cutting ants have to take great care of their garden fungus removing any old and suspicious material, such as introduced spores and bacteria, to waste chambers in order to avoid infection of the garden fungus. The survival of the whole ant colony depends on a healthy garden fungus [5,6,7]. The garden fungus can be devastated by pathogens such as *Escovopsis* (Hypocreales: Hypocreaceae) [6,8,9], *Fusarium* (Hypocreales: Nectriaceae) [10], *Syncephalastrum* (Mucorales: Syncephalastraceae) [10] or *Trichoderma* (Hypocreales: Hypocreaceae) [10]. In particular, the specialized fungal pathogen *Escovopsis weberi* (Muchovej and Della Lucia 1990, Hypocreales: Hypocreaceae) can destroy the ants’ colonies [6,9] using toxins targeting *L. gongylophorus* and the leaf-cutting ants [11,12,13]. Therefore, leaf-cutting ants make use of both mechanical cleaning and chemical treatment with metapleural gland secretions, to protect *L. gongylophorus* as well as to stimulate its growth [14,15,16,17,18].

In 1999, Currie et al. discovered that antibiotic-producing Actinobacteria, *Pseudonocardia* (Pseudonocardiales: Pseudonocardiaceae), growing on the integument of leaf-cutting ants in biofilms, support the ants as mutualistic symbiotic partners to fight infections [19]. In the following years it turned out that the microcosmos of leaf-cutting ants is much more complex than the interactions between a leaf-cutting ant species, its garden fungus, its *Pseudonocardia* symbiont and the pathogen *E. weberi.* Many microorganisms are found in the leaf-cutting ants’ microcosomos fulfilling different ecological functions: there are pathogens [5,6,9,10], as well as microbial symbionts protecting and supporting the growth of the garden fungus [19,20,21,22,23,24,25].

With the identification of the antifungal compounds from leaf-cutting ant symbionts, we started to gain an in-depth picture on the chemistry of the interspecies interactions in the microcosmos of leaf-cutting ants. In the last decade, a growing variety of polyketide and non-ribosomal peptide antifungals, e.g., candicidin [22], dentigerumycin [23,26], nystatin P1 [25], antimycins [27,28] or attinimycins [29], have been characterized from microbial symbionts of different leaf-cutting ant species indicating the diversity of defense mechanisms. Some of the antifungals appear to be common and widespread among symbionts of leaf-cutting ants and even occur at different geographical locations [22,25,29]. Some secondary metabolites from microbial symbionts of leaf-cutting ants were directly identified in their biological context (e.g., on the ants’ integument or in waste material) thus strongly suggesting their ecological function [27,29]. So far, all antimicrobial compounds from microbial symbionts were—as expected—complex secondary metabolites of polyketide or non-ribosomal peptide origin.

Among the isolated Actinomycetes from the leaf-cutting ant *Acromyrmex volcanus* (Wheeler 1937, Formicidae: Myrmicinae) we had obtained *Streptomyces* sp. Av25_4 [22]. This strain exhibited pronounced activity against *E. weberi*. Here, we describe the simple but efficient strategy *Streptomyces* sp. Av25_4 and other *Streptomyces* symbionts use in addition to complex secondary metabolites to control the fungal pathogen *Escovopsis*.

## 2. Materials and Methods

### 2.1. Chemicals

Activated charcoal, agar agar Kobe I, and mannitol were purchased from Carl Roth (Karlsruhe, Germany). Soy flour (Hensel Bio-Voll-Soja) was from W. Schoenenberger GmBH Co. KG (Magstadt, Germany). HPLC grade methanol was purchased from VWR International GmbH (Bruchsal, Germany). Ethyl acetate was distilled prior to use. For preparation of media, double distilled water was used. All other chemicals were from Sigma-Aldrich (Taufkirchen, Germany). Petri dishes were purchased from Greiner Bio-One GmbH (Frickenhausen, Germany).

### 2.2. Streptomyces Symbionts, Fungal Strains and Cultivation Conditions

*Streptomyces* strains were isolated previously from *A. volcanus* and *Acromyrmex echinatior* (Forel 1899, Formicidae: Myrmicinae) [22]. *Streptomyces* strains were cultivated on soy flour mannitol (SFM) agar plates (20 g soy flour, 20 g mannitol, 20 g agar 1 l ddH_2_O) [30] at 28 °C. *E. weberi* CBS 110660 and *Escovopsis aspergilloides* (Seifert, Samson and Chapela 1995, Hypocreales: Hypocreaceae) CBS423.93 were obtained from the Westerdijk Fungal Biodiversity Institute (Utrecht, the Netherlands). *Fusarium equiseti* FSU5459 (Corda Sacc. 1886, Hypocreales: Nectriaceae) was from the Jena Microbial Resource Collection (Jena, Germany).

*E. weberi*, *E. aspergilloides*, and *F. equiseti* were cultivated at 28 °C on either SFM agar or potato dextrose agar (PDA) plates. *L. gongylophorus* was isolated from an *Acromyrmex octospinosus* (Reich 1793, Formicidae: Myrmicinae) nest maintained at University of Kaiserslautern by Dr. Rainer Wirth. *L. gongylophorus* was cultivated on potato dextrose agar medium (PDA) or SFM agar plates at 30 °C, at ca. 40–50% humidity in the dark.

### 2.3. Screening Streptomyces Symbionts for Antifungal Volatile Compounds

Eight *Streptomyces* isolates [22] (Appendix A) from leaf-cutting ants were screened for their ability to produce antifungal volatiles using three compartment Petri dishes divided by a plastic barrier.

A *Streptomyces* symbiont was spread onto SFM medium in one compartment. The plates were wrapped with parafilm in order to avoid potential contamination with volatiles from the other cultures in the incubator and incubated at 28 °C. After 3 d, 80 µL of an *E. weberi* spore suspension was spread uniformly on the two remaining compartments of the divided plate containing SFM medium, and plates were again sealed with parafilm. As control, *E. weberi* was grown alone in the same way. The growth of *E. weberi* was monitored 5 d after inoculation of *E. weberi*. All bioassays were performed in triplicate. *Streptomyces* sp. Av25_4 inhibited the growth of *E. weberi* mycelium completely and was therefore selected for detailed investigations. In addition, it was tested whether *Streptomyces* sp. Av25_4 inhibits the growth of *E. aspergilloides* in the same way.

### 2.4. Trapping Volatiles Released by Streptomyces sp. Av25_4 and Evaluation of the Consequences on E. weberi Growth

Three compartment Petri dishes were used for the experiment (Figure 1, Appendix A). As a positive control, *Streptomyces* sp. Av25_4 was spread onto SFM agar in one compartment, and *E. weberi* was introduced as an agar plug (6 mm diameter) to PDA agar in a second compartment. The agar plug originated from a PDA plate of actively growing *E. weberi*. The third compartment was left empty.

For the volatile trapping experiment, the third compartment was filled with 2 g of activated charcoal in order to absorb the volatiles released by *Streptomyces* sp. Av25_4 and investigate their effect on the growth of *E. weberi*. The Petri dishes were sealed with parafilm after inoculation with *E. weberi* and further incubated for 5 d at 28 °C (Appendix A). As negative control, *E. weberi* was grown alone in the compartment containing PDA medium. The area of fungal growth was calculated by using image J 1.52a. Each bioassay was at least performed in triplicate.

### 2.5. Closed Loop Stripping (CLS) of Volatiles: GC-MS Analysis and Bioassays with the Collected Organic Volatiles

*Streptomyces* sp. Av25_4 was cultivated on SFM agar plates at 28 °C for 7d. The emitted volatiles were then sampled from *Streptomyces* sp. Av25_4. Through two holes in the side of the agar plate a closed-loop stripping pump (3V power supply, Fürgut, Tannheim, Germany,) fitted with an activated charcoal filter (1.5 mg, CLSA, le Ruisseau de Montbrun, Daumazan sur Arize, France) was inserted into the head space and the volatiles were collected for 1 h (Appendix A) [31]. The collected volatiles were eluted from the charcoal filter with 3 × 25 μL ethyl acetate. The extract was analyzed by GC-MS using a Trace GC Ultra hyphenated with an ISQ quadrupole mass spectrometer (Thermofisher, Dreieich, Germany). Separation was performed with an Optima 5 MS column (30 m × 0.25 mm, film 0.25 um, Macherey-Nagel, Düren, Germany). GC-MS conditions: inlet temperature 280 °C, splitless injection, hydrogen served as a carrier gas (0.9 mL/min). The volatile metabolite profiles were recorded using the following temperature programme: 50 °C isotherm for 3 min, 6 °C/min to 200 °C, 20 °C/min to 260 °C, 260 °C isotherm for 1 min. The mass spectrometer was operated in electron impact positive ionization mode at 70 eV. Compounds were identified by analysis of their mass spectra and comparison of their retention times as well as mass spectra to authentic reference compounds.

Volatiles collected (from days 4 to 9 for 24 h each) by closed loop stripping from 10 agar plates of *Streptomyces* sp. Av25_4 in a 2.5 L desiccator were tested for their inhibitory potential against *E. weberi*. The volatiles were eluted from the charcoal filters with 120 μL ethylacetate. In three compartment Petri dishes *E. weberi* was cultivated for 5 d. In the second compartment, a filter paper (1.5 cm diameter) was added. In total, 20 μL of the extract or ethylacetate as control was applied daily to the filter paper in the three compartment Petri dish bioassay. After 10 d the growth of *E. weberi* was analyzed by determining the covered area using image J 1.52a.

### 2.6. Change of the pH upon Growth of Streptomyces sp. Av25_4

The change of the pH in the growth medium was monitored using phenol red as pH indicator [32]. *Streptomyces* sp. Av25_4 was spread on the SFM medium of the first compartment. The other two compartments contained SFM medium with 0.002% of phenol red as pH indicator. The plates were sealed with parafilm and the color change in the medium was recorded by taking pictures at 4, 7, and 10 d after incubation.

### 2.7. Detection of Ammonia

Whether *Streptomyces* sp. Av25_4 produces ammonia or not was investigated by using *ortho*-phathaldialdehyde (OPA) derivatization and fluorescence detection of the resulting derivative [33] as well as by ^1^H NMR analysis [34]. The OPA derivatization reagent was prepared by mixing 10.5 mg OPA dissolved in 210 μL methanol and 10.5 mg of sodium sulfite dissolved in 630 μL of sodium phosphate buffer (100 mM, pH adjusted to 12.0 with 2M NaOH). *Streptomyces* sp. Av25_4 was grown for 7 d at 28 °C on SFM agar plates. Ten plates with fully-grown *Streptomyces* sp. Av25_4 were transferred into a desiccator (2.5 L) that was closed with a septum. A CLS pump was used to directly transfer the volatiles released within 1 h by *Streptomyces* sp. Av25_4 from the desiccator into the OPA derivatization solution (20 μL of OPA reagent in 500 μL of phosphate buffer) in a 4 mL glass vial.

Similarly, a CLS pump was used to directly transfer the volatiles released within 3 d by 6 d grown *Streptomyces* sp. Av25_4 from the desiccator (size 2.5 L, 10 SFM agar plates) into the 250 mM HCl solution (500 μL prepared in ddH_2_O) in a 4 mL glass vial. DMSO-d6 (50 μL) was added for calibration of the NMR spectrometer. ^1^H-NMR spectra were recorded for the obtained solution of the collected volatiles, a solvent control and an equally prepared ammonium standard using a Bruker Avance III 600 spectrometer (Bruker BioSpin GmbH, Rheinstetten, Germany) with water suppression. Data was recorded using the Bruker Topspin software. Data was analyzed using MestReNova 14 software (Mestrelab Research, Santiago de Compostela, Spain).

### 2.8. Quantification of Ammonia

From the *E. weberi*/*Streptomyces* sp. Av25_4 co-culture bioassays, where the strains were separated by a plastic barrier using the three compartment plates (see above), an agar plug (1 × 1 cm area) was taken from the PDA agar compartment in which *E. weberi* was inoculated to quantify ammonia. The agar plugs were collected into 2 mL Eppendorf tubes at different time points (2, 4, 6, 8, 10, and 12d) and stored at −20 °C. After freezing and thawing of the samples, the released liquid was collected by centrifugation at 16,873 g for 3 min. Then, 20 μL of the obtained liquid was diluted with 500 μL of 0.1 M phosphate buffer (pH 12.0) and derivatized with 20 µL of OPA reagent. The samples were kept in the dark for 30 min at room temperature and analyzed using a Dionex Ultimate 3000 UHPLC system connected to a Dionex RF 2000 fluorescence detector. UHPLC method: HPLC column: Thermo Scientific Accucore RP-MS column: 150 × 2.1, 1.5 μm); solvent A: H_2_O 0.1% acetic acid, solvent B: MeOH 0.1% acetic acid; gradient elution: 35–100% B in 20 min, 5 min 100% B, flow rate: 0.25 mL/min.

The fluorescence intensity of thus formed fluorescent product was measured at emission wavelength of 420 nm after excitation at 360 nm [33].

Ammonia was quantified by comparison of the peak areas to an external calibration curve (2.5, 5, 7.5, 10 mM ammonia). All values where at least determined in triplicate.

Similarly, the amount of ammonia that accumulated in the PDA agar compartment when 5 mL ammonia (5, 7, 9, and 11 mM) was supplied into one compartment without medium of the three compartment agar plate was determined. Number of biological replicates per data point was four.

### 2.9. Influence of Ammonia on the Growth of E. weberi, E. aspergilloides, F. equiseti, and L. gongylophorus

In order to study how ammonia affects fungal growth, three or two compartment Petri dishes were used. The first compartment containing PDA medium was inoculated with an *E. weberi*, *E. aspergilloides, F. equiseti* fungal hyphae/spore solution or an *L. gongylophorus* agar plug (6 mm or 1.5 cm diameter). The second compartment was supplemented with 5 mL of 5, 7, 9, 11, 15 mM or 10 mM (in case of *L. gongylophorus*) ammonia solution and the third compartment was left empty. The Petri dishes with *E. weberi, E. aspergilloides*, or *F. equiseti* were sealed with parafilm and further incubated for 5–10 d at 28 °C. In case of *L. gongylophorus* the plates were incubated for 3 weeks at 28 °C and 40–50% humidity.

For control plates, sterile water was added instead of the ammonia solution. Each concentration was at least tested three times. The area of fungal growth was calculated by using image J 1.52a.

### 2.10. Influence of Medium Alkalinization on the Growth of E. weberi

The growth of *E. weberi* on SFM agar plates at pH 7.5, 8.0, and 8.5 was monitored. The SFM agar plates were inoculated with an *E. weberi* agar plug (6 mm diameter). The pH of SFM medium was adjusted by using 3M sodium hydroxide solution. The growth of *E. weberi* was analyzed after 5 d of incubation. Each pH was at least reproduced three times.

### 2.11. Determination of the pH and Quantification of Ammonia from Waste of Leaf-Cutting Ants

In order to determine the pH of waste from different *Atta* and *Acromyrmex* leaf-cutting ant nests (Appendix A) 2 g of waste was suspended in 20 mL double distilled water (pH7).

The pH was determined with a pH electrode after 3 min.

The presence and amount of ammonia in waste of leaf-cutting ants was determined by collecting ammonia with molecular sieves (0.5 nm pore size, Merck, Darmstadt, Germany) and comparing the values to an ammonia reference curve. Molecular sieves (0.5 g) were added into a 4 mL glass vial that was posed into a 50 mL flask with 2 g of waste or 2 g of an (2.5, 5, 10 mM) ammonia solution. Then the flask was closed. After 16 h, the molecular sieve was collected and ammonia was eluted with 1 mL of 250 mM HCl and ultrasonication for 10 min. The solution was centrifuged at 16,873 g for 1 min and the supernatant was collected. Ammonia was quantified using the indophenol-blue method adapted from the DIN 38406–5norm (DIN-Normenausschuss Wasserwesen 1983). Then, 50 μL of solution A (0.65 g sodium salicylate, 0.65 g trisodium citrate dihydrate and 10 mg sodium nitroprusside dihydrate in 5 mL ddH_2_O) and 50μL of solution B (10 mg sodium dichloroisocyanurate in 5 mL NaOH (32 g L^−1^)) were added to a 1/10 dilution of 500 μL molecular sieve extracts of the ammonium standards or the molecular-sieve extracts from waste. Reaction mixtures were incubated for 3 h in the dark. The absorption was measured at 655 nm and the amount of ammonia was calculated based on the ammonia reference values. The measurements were performed in triplicate [34].

## 3. Results

### 3.1. Streptomyces sp. Av25_4 Produces Antifungal Volatiles

As *Streptomyces* strains are well known to produce a large variety of organic volatile compounds [35,36,37] we screened *Streptomyces* symbionts from leaf-cutting ants for whether the production of volatile compounds inhibits the growth of the pathogenic fungus. *E. weberi* and the *Streptomyces* isolates were cultivated in three compartment Petri dishes so that they grow physically separated from each other but can exchange volatile compounds (Figure 1). The volatile compounds of 3 out of 8 strains (*Streptomyces* sp. Av25_4, *Streptomyces* sp. Av26_5, *Streptomyces* sp. Av28_3) clearly inhibited *E. weberi* growth (Appendix A, Appendix A). In particular, *Streptomyces* sp. Av25_4, nearly completely inhibited the growth of *E. weberi* (Figure 1). Therefore, *Streptomyces* sp. Av25_4 was selected in order to reveal which compound(s) is/are responsible for the observed antifungal activity. By adding activated charcoal to the assay to adsorb any volatiles produced, we observed a clear decrease of the antifungal effect against *E. weberi*, leading to 42 ± 11% increase in the area covered by *E. weberi* compared to the control (Figure 2, Appendix A). Notably, exposure to less ammonia due to the presence of activated charcoal (Figure 2D) still led to a change in pigmentation compared to the control (Figure 2B).

### 3.2. Analysis of Volatiles from Streptomyces sp. Av25_4

In order to study the volatile compounds released by *Streptomyces* sp. Av25_4, the volatile compounds were collected using closed loop stripping (Appendix A). GC-MS analysis revealed a range of organic compounds, such as alcohols, ketones, aldehydes, terpenes, and lactones. Five compounds, namely geosmin, 2-methyl-isoborneol, 2-phenylethanol, nonanal, and benzaldehyde were selected because they were both abundant and some of them we suspected to exert antifungal activity against *E. weberi* (Appendix A). Among the five compounds tested, only nonanal and benzaldehyde inhibited *E. weberi* growth at high concentrations and the aldehydes were also present in the SFM medium control. Thus, these compounds cannot account for the observed complete growth inhibition of *E. weberi* by volatiles from *Streptomyces* sp. Av25_4.

Moreover, organic volatiles collected for 24 h by closed loop stripping from 10 *Streptomyces* sp. Av25_4 plates and applied daily to the *E. weberi* growth assay in a two compartment Petri dish did not show any effect on the growth of *E. weberi (*Appendix A).

### 3.3. Streptomyces sp. Av25_4 Overproduces Ammonia

As the organic volatile extract from the closed loop stripping of *Streptomyces* sp. Av25_4 as well as the selected identified volatile organic compounds did not show activity against *E. weberi*, we suspected that *Streptomyces* sp. Av25_4 releases another potent antifungal volatile. Since our GC-MS approach was only suitable for nonpolar organic compounds, we suspected that polar volatiles may confer the observed antifungal activity.

Using the three compartment Petri dish we monitored whether volatiles from *Streptomyces* sp. Av25_4 alter the pH of the *E. weberi* PDA growth medium that was supplemented with the pH indicator phenol red [32]. Upon alkalinization phenol red changes its color form pale yellow (pH 1–7.3) to purple (pH > 7.3) [32]. At the beginning of the experiment, the pH increase was observed near *Streptomyces* sp. Av25_4. However, after 7 d the whole *E. weberi* growth medium appeared entirely pink, indicating the alkalinization is caused by a volatile compound from *Streptomyces* sp. Av25_4 (Appendix A).

Some *Streptomyces* strains can produce volatile amines such as ammonia [32,34] and trimethylamine [38]. Such volatile amines can alter the pH in the physically separated compartment with *E. weberi*. Therefore, we investigated if *Streptomyces* sp. Av25_4 produces volatile amines. The volatile amines were trapped in 250 mM HCl solution using the closed loop stripping pump (Appendix A). The collected sample was analyzed by ^1^H-NMR spectroscopy using water suppression. The ^1^H-NMR spectrum revealed an intense triplet signal at 7.01 ppm with a coupling constant of 51.6 Hz, proving that *Streptomyces* sp. Av25_4 produces a high amount of ammonia (Appendix A) [34]. Signals accounting for other volatile compounds were only detected in traces in the ^1^H-NMR spectrum.

Furthermore, the production of ammonia by *Streptomyces* sp. Av25_4 was quantified by derivatization with *ortho*-phthalaldehyde (OPA), HPLC separation and fluorescence detection (Appendix A) [19]. *Streptomyces* sp. Av25_4 emitted ammonia in large quantities (up to ca. 8 mM) in comparison to other symbionts (Figure 3). Trimethylamine was not detected in the volatile blend of *Streptomyces* sp. Av25_4.

### 3.4. Ammonia from Streptomyces sp. Av25_4 Strongly Inhibits the Growth of Escovopsis

The growth inhibitory effect of ammonia against *E. weberi* was assessed by adding 5 mL of ammonia solutions (5, 7, 9, and 11 mM) in one compartment of a three compartment plate (Appendix A). The volatile ammonia received in the *E. weberi* compartment of an ammonia concentration of 5, 7, and 9 mM, respectively, inhibited almost 37 ± 9%, 56 ± 4%, and 77 ± 2% of the *E. weberi* growth (Appendix A). The 11 mM ammonia completely inhibited the growth of *E. weberi*. Additionally, *E. aspergilloides* was strongly inhibited by ammonia (15 mM) (Appendix A). However, ammonia (10 mM) neither affected the growth of *F. equiseti* nor of *L. gongylophorus* (Appendix A).

In order to assess whether or not *Streptomyces* sp. Av25_4 releases sufficient volatile ammonia to inhibit *E. weberi*, the concentration of received volatile ammonia in the PDA agar compartment with *E. weberi* was quantified after 2, 4, 6, 8, 10, and 12 d of *Streptomyces* sp. Av25_4 growth.

The ammonia concentration in PDA agar extracts of the *E. weberi* growth compartment was quantified by OPA derivatization and fluorescence detection in comparison to a standard ammonia reference curve (y = 3136.8x + 3578.6, R^2^ = 0.999). After 2 d of growth, *Streptomyces* sp. Av25_4 caused an accumulation of ca. 4 mM ammonia in the PDA growth medium (Figure 3). After 10 d, the concentration of ammonia further increased to almost 8 mM. The ammonia concentrations of the PDA agar extract of the *Streptomyces* sp. Av25_4/*E. weberi* co-culture match to the concentration necessary to almost completely inhibit the growth of *E. weberi*.

### 3.5. Alkalinization of the Medium and Growth Inhibition

As the base ammonia was found to be sufficient to explain the observed growth inhibition of *E. weberi*, we studied if the alkalinization of the medium is the reason for the growth inhibition. Therefore, the growth of *E. weberi* was monitored at pH 7.5, 8.0, and 8.5 adjusted with sodium hydroxide instead of ammonia. *E. weberi* was 90 ± 5% inhibited at pH 7.5 (Appendix A). At pH 8.0, the growth of *E. weberi* was completely inhibited.

### 3.6. Waste of Leaf-cutting Ants Can Be Strongly Basic

Since the nests of *A. volcanus* and *A. echinatior* ants we previously isolated our ammonia producing *Streptomyces* strains from, in particular *Streptomyces* sp. Av25_4 (see above, Appendix A), were unfortunately not available to us any more, we investigated whether other nests of leaf-cutting ants exhibit a basic pH in their waste and produce ammonia that strongly inhibits the growth of *Escovopsis*. The pH of waste samples from 19 *Atta* and *Acromyrmex* colonies was determined (Appendix A). Six out of 19 waste samples exhibited a basic pH of > 8. Interestingly, the pH of waste of colonies of the same leaf-cutting ant species varied and could be either acidic or basic. However, the pH of the individual nests remained throughout our experiments. All basic waste samples contained ammonia, as determined by collecting ammonia with molecular sieves followed by quantification using the Berthelot reaction [34]. The waste of *Acromyrmex ambiguus* (Emery 1888, Formicidae: Myrmicinae) colony L contained ammonia in high amounts (8 mM) that strongly inhibited *Escovopsis* growth.

## 4. Discussion

Microorganisms, in particular *Actinomycetes*, often produce a large variety of volatile compounds [35,36,37]. However, in contrast to volatiles from plants and animals, the biological function of microbial volatiles, in particular in the ecological context, remained in most cases unknown. As volatiles can diffuse over long distances, they are ideally suited for communication as well as deterrence of competitors [35]. For some microbial volatile compounds, e.g., the terpene albaflavenone [39] antibiotic activity was established. Studying the biological role of volatile compounds from *Actinomycetes* recently gained attention among researchers interested in the chemical ecology of *Actinomycetes* [35,36,37]. Huang et al. revealed that *Solenopsis invicta* ants favor soils rich in *Actinomyces* bacteria by recognizing their characteristic volatiles, 2-methylisoborneol and geosmin [40]. In the context of leaf-cutting ants, Silva-Junior et al. recently discovered that *Actinomyces* symbionts produce the same pyrazines [41] that the ants use as infochemicals [42,43,44]. These findings nicely demonstrate that volatiles should be investigated for their ecological role in interspecies interactions between ants and their microbial symbionts.

Now we demonstrate that the microbial symbiont of *A. volcanus* ants, *Streptomyces* sp. Av25_4, overproduces ammonia up to 8 mM. Moreover, also other *Streptomyces* symbionts from leaf-cutting ants we investigated produce ammonia, however, less pronounced than *Streptomyces* sp. Av25_4 (Appendix A, Appendix A). Microbial symbionts from *A. volcanus* and *A. echinatior* produce ammonia, indicating that ammonia production is common among *Streptomyces* symbionts of leaf-cutting ants.

The 8 mM of ammonia (Figure 3), as produced by *Streptomyces* sp. Av25_4, strongly inhibited the growth of the pathogen *E. weberi*, explaining the observed inhibition of *E. weberi* in co-culture with *Streptomyces* sp. Av25_4 separated by a physical barrier. Our results are perfectly in line with the recent, detailed mechanistic study by Avalos et al. [32], which nicely demonstrated that many *Streptomyces* strains produce ammonia that can serve them to inhibit potential bacterial competitors. Moreover, volatile amines play important physiological roles in some *Streptomyces* strains as trimethylamine was found to induce exploratory growth [38], while ammonia induces droplet production [34], and exerts antimicrobial [32] and anti-oomycetal effect [45]. In addition, ammonia can potentiate the activity of antibiotic secondary metabolites [32]. Both tested *Escovopsis* strains, *E. weberi* and *E. aspergilloides*, were sensitive to ammonia whereas the garden fungus *L. gongylophorus* and *F. equiseti* were not affected by ammonia rendering the defense by ammonia highly suitable to fight the specialized pathogen of leaf-cutting ants *Escovopsis*.

However, the fungus garden of leaf-cutting ants appears to be maintained at an acidic pH of 5.2 [46,47]. In order to avoid infections by *Escovopsis* and other pathogens, leaf-cutting ants carefully weed their garden fungus, remove any old and suspicious material containing pathogens from their garden fungus and deposit the waste in special waste chambers. Thus, the outbreak of infections in the nest of leaf-cutting ants can mostly be kept under control [15,17,18]. In order to avoid reintroduction of pathogens from the waste sites by leaf-cutting ants performing the waste management back into the nest efficient disinfection of the waste is required. Thus, we reasoned whether the ammonia from microbial symbionts may contribute to inactivate the waste material of leaf-cutting ants.

Indeed, the waste of 6 out of 19 (Appendix A) ant nests exhibited a pH of ca. 8, that we showed to strongly inhibit *Escovopsis* growth (Appendix A). With the example of the *A. ambiguus* nest colony L we proved that the basic pH is caused by ammonia. Alternatively, we observed a pH around 5 from 6 out of 19 leaf-cutting ant nests tested, indicating that both acidification and alkalization are used as an efficient and simple strategy to disinfect the waste. Indeed, the use of acids or bases is an efficient and common way of microorganisms to outcompete competitors, that humans make use of e.g., for preservation of food [48,49].

Interestingly the use of acids or bases is not species specific but rather specific for the individual nest potentially suggesting that it may depend on the acquired symbionts.

Future experiments are needed to study in-depth which antiinfective strategies are used to inactivate dangerous pathogens in the waste dumps of different leaf-cutting ant species and nests. In order to reveal the functioning of the microcosomos of leaf-cutting ants it is crucial to identify and consider the whole variety of low and high molecular weight compounds present. Our experiments demonstrate that one strategy to inhibit *Escovopsis* is to maintain a basic pH (Figure 3) using the volatile base ammonia. Compared to the complex secondary metabolites of *Actinomyces* symbionts of leaf-cutting ants, e.g., the polyketide candicidin [22], ammonia is a very simple but remarkably efficient compound. As established by Avalos et al., ammonia production can be a highly efficient way to outcompete competitors because of its biosynthetic simplicity and thus metabolically low cost of production. *Streptomycetes* often produce ammonia by degradation of amino acids [32]. Ammonia production only requires few enzymes in contrast to the mega-synthases needed for polyketide or non-ribosomal peptide production [32].

For the first time we demonstrated that in the microcosmos of leaf-cutting ants, microbial symbionts, such as *Streptomyces* sp. Av25_4, overproduce ammonia that strongly inhibits the growth of the fungal pathogen *Escovopsis*. With simple cost-effective ammonia, *Streptomyces* strains contribute to render the waste of leaf-cutting ants basic to discourage the growth and spreading of *Escovopsis* from their waste chambers.

## Figures and Tables

**Figure 1 microorganisms-09-01622-f001:**
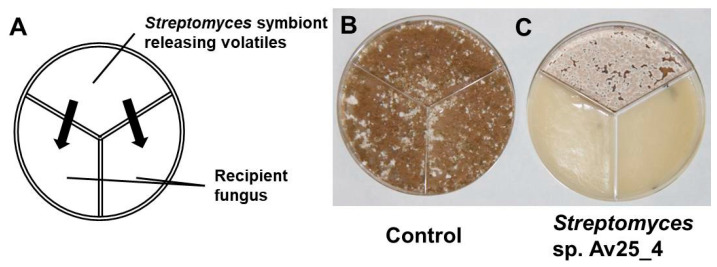
Screening *Streptomyces* symbionts of leaf-cutting ants for volatile antifungal compounds that inhibit *E. weberi*. (**A**) Scheme of the experimental set-up in a three compartment Petri dish. (**B**) Growth of *E. weberi* (control), (**C**) volatiles emitted by *Streptomyces* sp. Av25_4 completely inhibited the growth of *E. weberi*.

**Figure 2 microorganisms-09-01622-f002:**
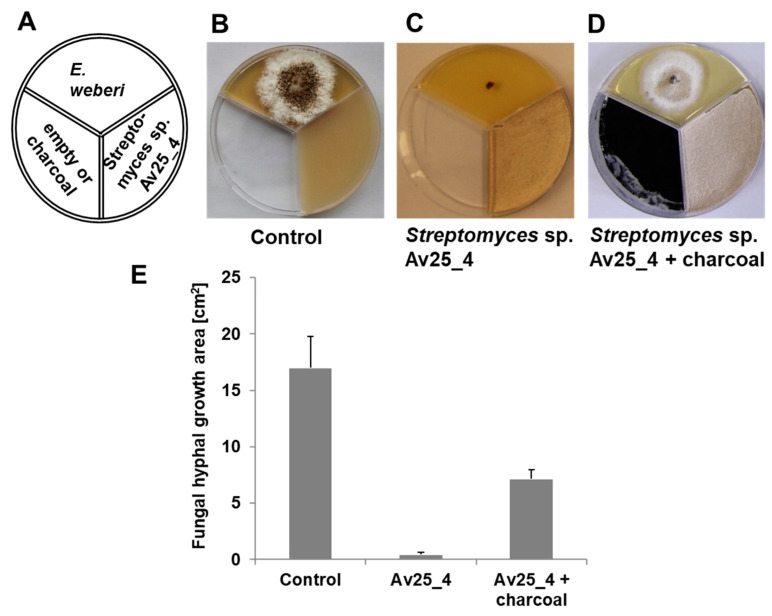
Antifungal activity of volatiles produced by *Streptomyces* sp. Av25_4 against *E. weberi* with and without activated charcoal in the third compartment. (**A**) Scheme of the experimental setup. (**B**) *E. weberi* grown for 4d in the first compartment. (**C**) *E. weberi* grown for 4d in first compartment and *Streptomyces* sp. Av25_4 grown for 7d in the second compartment. (**D**) *E. weberi* grown for 4 d in the first compartment, *Streptomyces* sp. Av25_4 grown for 7d in the second compartment and 2 g activated charcoal in the third compartment. (**E**) Quantification of the *E. weberi* growth area analyzed by image J.

**Figure 3 microorganisms-09-01622-f003:**
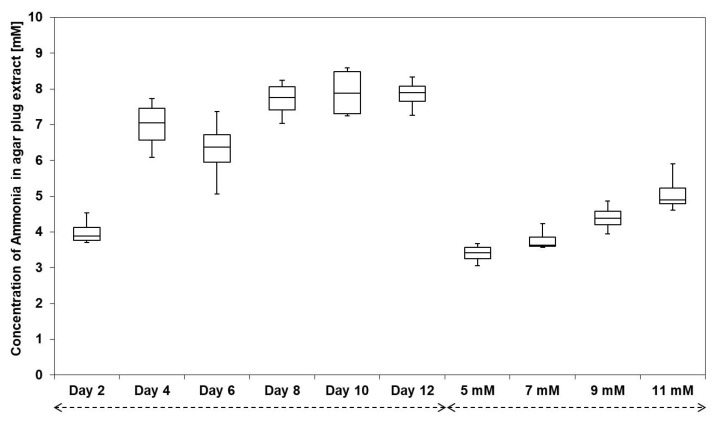
Box plot diagram showing the amount of ammonia released by *Streptomyces* sp. Av25_4 after 2, 4, 6, 8, 10, and 12 d in the agar plug of the PDA medium of the *E. weberi* compartment. Ammonia in the PDA agar plug of the *E. weberi* compartment after supplementation of ammonia (5, 7, 9, and 11 mM). The whiskers represent the minimum and maximum values and the segment inside the rectangle shows the median values (four biological replicates).

## Data Availability

Data is presented in the paper and the Appendix A. Original data files will be available upon request from the authors.

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
