# Peer review of "Ammonia Production by *Streptomyces* Symbionts of *Acromyrmex* Leaf-Cutting Ants Strongly Inhibits the Fungal Pathogen *Escovopsis"

_microorganisms, 2021, doi:10.3390/microorganisms9081622_

Round 1

Reviewer 1 Report

Just a note on biological nomenclature code: all the scientific names (fungi, insects, etc..) when reported for the first time in the text, should be written in full with Authority and systematics. Moreover the italics should be used only for Genus and species

Author Response

Dear reviewers,

Dear editor,

Thank you for reviewing our manuscript and your comments to improve our paper.

We followed all your suggestions.

Best wishes,

Dieter Spiteller

Reviewer 2 Report

This study by Dhodary and Spiteller shows that volatile ammonia produced by Streptomyces, a symbiont of leaf-cutting ant Acromyrmex inhibits the growth of pathogenic fungus Escovopsis.  The article is written in a concise manner, thus easy to read and understand. The experiment is properly designed and well-executed.

Major comment

The introduction is very short. Having some details in the introduction regarding chemical warfare between ant-symbionts and pathogens would benefit the readers.

Minor comments

Line 259, 322, 334, 363 – Check the placing of punctuation marks (.).

Please review the article again thoroughly for the error.

Line 356 – Is a preposition missing? – “not affected by ammonia”

Author Response

(The authors gave the same response as above.)
